# Clarification of Most Relevant Concepts Related to the Microalgae Production Sector

**Vítor Verdelho Vieira [1,2], Jean-Paul Cadoret [1,3], F. Gabriel Acien [4,]\* and John Benemann [5]**

[1] European Algae Biomass Association, Viale Belfiore, 10, 50144 Florence, Italy; Vitor.verdelho@eaba-association.org (V.V.V.); jean-paul.cadoret@eaba-association.org (J.-P.C.)
[2] A4F—Algae for Future, Rua Eng. Clément Dumoulin Business Park, 2625-106 Póvoa de Santa Iria, Portugal
[3] ALGAMA, 81 rue Réaumur, 75002 Paris, France
[4] Department of Chemical Engineering, University of Almería, 04120 Almería, Spain
[5] MicroBio Engineering Inc., San Luis Obispo, CA 93406, USA; johnbenemann@microbioengineering.com
[*] Correspondence: facien@ual.es

**Abstract:** Microalgae (including cyanobacteria) are the basis for an emerging worldwide industry but still face significant bottlenecks in contributing to the global economy. It is an enormous challenge to translate experiences from established industries such as aquaculture and agriculture to the microalgae sector. In particular, this includes the challenge of adapting regulations that apply to such macroscopic production and mindsets, to the microscopic world of microalgae and to the scale-up to a million times smaller. Current European and country-based regulations do not always, indeed rarely, consider relevant specific issues that limit the path for innovation and growth applicable to the microalgae sector. In this work, the boundaries for the main issues impacting this sector are presented and discussed. Examples and possible analytical frameworks are presented in a question and answer format. Relevant key topics and related boundaries are discussed: What are algae and how do microalgae differ from macroalgae? Why are algae and specifically microalgae relevant? Is algae cultivation an aquaculture process? Can algae and specifically microalgae be classified as vegetables and their production be classified as agriculture or are they an industrial process? How is algaculture compared with other agricultural sectors? What are organic algae? Can microalgae be grown in wastewater and how can they be used? What are toxic algae? What are the bottlenecks for microalgae culture scale-up? How does the microalgae biodiversity contribute to their development? We conclude that microalgae are developing as a novel agricultural enterprise that can provide major benefits to a sustainable circular economy and environment but require appropriate regulations and support from governments and businesses, recognising its unique attributes and potential.

**Keywords:** algae; microalgae; aquaculture; biomass; bottlenecks; regulatory; innovation; bioeconomy

## 1. Introduction

Microalgae, including cyanobacteria, have been proposed as a solution to the problems of mankind, from food, feeds and biofuels production to wastewater treatment and greenhouse gases abatement, among others. Indeed, the applications of microalgae are potentially enormous, covering the entire spectrum of products and services including the following: (i) pharmaceuticals and cosmetics; (ii) high-quality food and nutraceuticals; (iii) commodity foods and feeds; (iv) chemicals, biostimulants and biopesticides; (v) biofertilizers and biofuels; and (vi) wastewater treatment and $CO_2$ removal from flue gases [1]. However, current commercial processes are still incipient when compared with well-established biotechnological industries based on higher plants, bacteria, yeast or fungi. The current microalgae production worldwide approaches 50.000 t/year, mainly for five species, *Spirulina*, *Chlorella*, *Dunaliella*, *Haematococcus* and *Nannochloropsis* [2]. Microalgae biomass currently produced is mainly devoted to human food and nutritional-related products, as this sector can cover the current biomass production cost, which can range

from about 5 to over 20 €/kg, using a variety of open or closed cultivation systems [3]. A major limiting factor is the currently small production capacity of most microalgae production facilities ranging from less than 1 ha to at most 100 ha, producing from a few tons to about 1000 t/year. In addition to the scale, the production cost of microalgae biomass varies, depending on several factors such as the production system, services (energy and manpower), location and certifications (e.g., organic production), and the market value depends in part on marketing and the stage in the value chain [2]. For *Chlorella* sp. and *Spirulina,* bulk market prices range between 25 and 70 €/kg. For *Dunaliella*, *Haematococcus* and *Nannochlorpsis*, these are used for the extraction of high-value consumer products (carotenoids and omega-3 fatty acids) are in a considerably higher production cost and price range, generally over 150 €/kg [2].

To enlarge the contribution of microalgae to the global and European bioeconomy, it is necessary to increase both their production capacity and the range of their applications and will require overcoming current bottlenecks, both technologically and regulatorily. Technologies are required that efficiently manage large volumes of water while maintaining the optional culture conditions required by microalgae strains. Integration with wastewater treatment and nutrient recovery processes can also increase the sustainability of algae production. Biological challenges are related to the improvement of strains already used, as well as novel strains, and control of contaminations, from pests, diseases, other algae, etc. [4]. The automation and optimization of technologies for biomass production are also required [5] as are biomass harvesting and processing schemes. Finally, improved business models are required to fully monetize the microalgae biomass, avoiding wastes while maximizing value chains to improve circularity and material efficiency [6]. However, current regulation can limit such further developments of the microalgae industry, as no regulations exist or only consider the utilization of specific microalgae strains and products [7]. Current uncertainties related to microalgae applications are a consequence of unclear regulatory boundaries.

This work aims to clarify some of these uncertainties and to provide a clear scenario for microalgae technologies and applications, in order to help de-risk investments in this sector and contribute to its expansion, beyond the current, very limited, nutritional and speciality applications, such as cosmetics. In the following sections, responses to the most relevant issues and uncertainties are provided in a question and answer format, also with inputs from members of the European Algae Biomass Association (EABA).

## 2. What Are Algae?

Algae are polyphyletic in origin (follow multiple and independent evolutionary lines; Figure 1). They are highly diverse with the common property that they carry out oxygenic photosynthesis, converting $CO_2$ to biomass and releasing $O_2$, although many can also grow in the dark on organic compounds, with some species even having lost their $O_2$ evolution capacity. They differ from higher plants, all of which evolved from green algae, one class of algae, by lacking vascular systems for nutrient transport. Phycology, the study of algae, developed historically as a discipline focused on the morphological, physiological and ecological similarities and diversity of these organisms. Microalgae comprise both eukaryotes (including five distinct evolutionary lineages) and prokaryotes (the cyanobacteria). Macroalgae or seaweeds differ mainly in size, with some growing up to 60 m in size and by growing in seawater. Some authors have recently started referring to larger freshwater filamentous microalgae as macroalgae.

Algae, unicellular, colonial or larger multicellular species, can be free-floating or attached to various surfaces, such as benthic algae growing on the bottom of streams or shallow lakes. Although seaweeds and most microalgae are strictly photoautotrophic, that is they grow on inorganic carbon sources ($CO_2$ or bicarbonate) and light energy, some microalgae species, as noted above, can also grow using organic compounds, in the light (mixotrophic growth) or the dark (heterotrophic growth), sometimes combining all three

nutrition modes. Herein, the focus is on both microscopic microalgae, including filamentous and attached species, as well as on the macroscopic seaweeds.

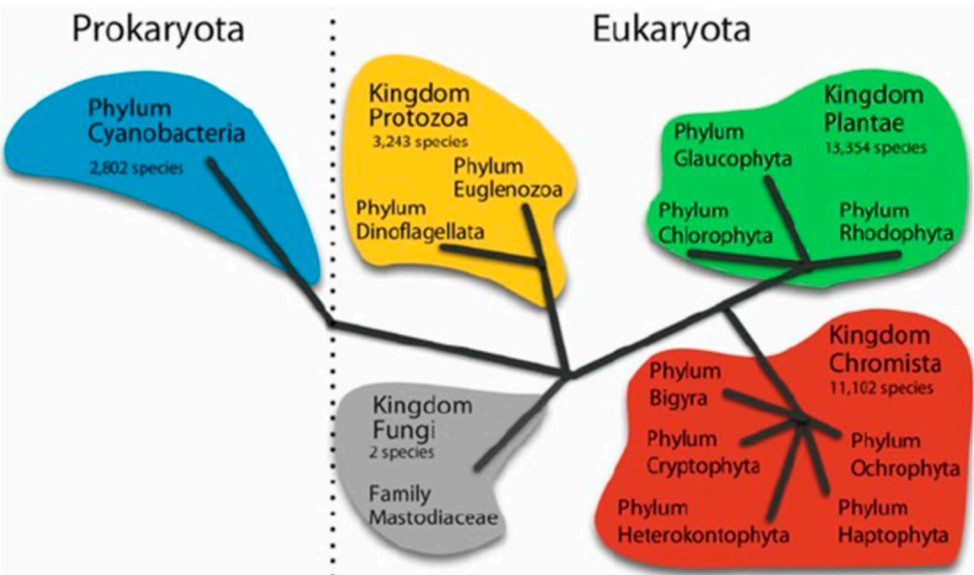

**Figure 1.** Distribution of algae among groups in the Tree of Life as recognized by the ITIS and Species 2000 (and ife.org) in 2011. The deep classification of algae is the subject of great debate, and even the higher clades have been discussed and revised recently [8–10].

Generally, microalgae, including cyanobacteria, are aquatic organisms, growing in fresh-, brackish- and saltwater, up to very high salinities. Some species can grow on rocks and other surfaces intermittently exposed to even very little moisture, and many microalgae species can survive for long periods, even hundreds of years, as resting stages or cysts, in extreme conditions, in soils, sediments, dry environments and low temperatures [11,12]. Seaweeds live strictly in seawater, and they can survive in littoral areas only if exposed to daily tides.

In brief, as the first boundary, algae is a common name for a wide range of aquatic organisms and includes both micro-and macroalgae.

### 3. Why Are Algae Important?

Cyanobacteria are responsible for the current life on the planet. Early in the history of life, cyanobacteria changed the planet's atmosphere by producing oxygen, thus paving the way for the evolution of eukaryotic organisms, including us. The oil we currently exploit comes mostly from Cretaceous deposits of marine algae. Today, algae are relevant, because they are the basis of food chains in oceans and freshwaters and because algae promise new crops. Microalgae as single-cell organisms can replicate in a few hours, making it possible to produce large amounts quickly. Microalgae also have a large diversity in components and composition, allowing for a wide range of applications, from nutritional products to animal feeds, biofertilizers, bioplastics, biofuels and much more. Microalgae already contribute to a wide range of sectors in the bioeconomy, with an increasingly social, economic and environmental impact, and are regulated in the European framework [7,13].

In brief, as the second boundary, algae are primary producers, in the first level of the food chain.

### 4. Is Algae Cultivation Aquaculture?

Aquaculture involves the breeding, rearing and harvesting of fish, shellfish, algae and other aquatic plants and animals, in brief, the farming of such organisms [14,15]. This

distinguishes aquaculture, also called mariculture, fish and shellfish farming or algaculture (for the cultivation of seaweeds and microalgae), from wild catch fishing.

Aquaculture requires the management of the production process, from the development of brood stocks to the raising of larval and juvenile stages, feeding and protection from predators, diseases, etc. It can involve the individual or corporate ownership and management of the production systems. For statistical purposes, aquatic organisms harvested by an individual or corporate body who owns them throughout their rearing period are counted as aquaculture, while aquatic organisms which are exploitable as a common property resource are included under fisheries [16,17].

> Is algae production an aquaculture process?

Yes. However, some algae are harvested from the environment, such as seaweeds collected on-shore or harvested off-shore, are classified as fisheries, not aquaculture.

In brief, as the third boundary, algaculture involves human effort in managing and growing the organisms involved. Algae cultivation always takes place in an aquatic environment. Algae applications in agriculture do not involve actual growth or cultivation.

### 5. Are Algae Plants and Therefore Vegetables?

Vegetables are plants that are consumed by humans or other animals as food. The original meaning of vegetable refers to all edible plant matter, including flowers, fruits, stems, leaves, roots and seeds. Algae are plants, based on the common usage of the term and their inclusion into the field of botany. The green algae or chlorophytes originated from an endosymbiotic event in which an ancestral non-photosynthetic microbe acquired a cyanobacterium (blue-green alga) are the ancestors of all higher and lower plants. The red algae resulted as well from such an initial endosymbiosis. Other types of algae, such as brown algae and diatoms, dinoflagellates, eustigmatophytes and euglenoids, resulted from additional endosymbiosis events of a red or green alga with other eukaryotic host cells. The common term algae include all the species that carry out oxygenic photosynthesis, from prokaryotes (the cyanobacteria) to several kingdoms of the eukaryotes.

As the fourth boundary, algae are plants, and therefore, they or their edible parts are vegetables.

### 6. Is Algaculture Agriculture?

Algae production can be considered as agriculture, defined as the art and science of growing plants and raising animals, for food, feeds and many other economic activities. Worldwide agriculture is undergoing rapid technological changes, driven by rapid advances and convergences in biotechnology, information technologies (e.g., precision agriculture), material sciences and others, driven by market and societal demands, changing all current farming practices. Crops can be classified into six categories based on their applications in foods, feeds, fibers, oils, ornamentals and industry. Algae fit in all categories and already are crops, with one-tenth of thousands of tons of microalgae being cultivated in large pond systems and millions of tons of seaweeds farmed in near-shore environments.

> Are algae a crop?

Yes. Algae are a crop with a diverse number of species being cultured.

> Is algae biomass different from the other crops?

No. Algae biomass is not substantially different from other crops; its biochemical composition and nutritional properties are analogous to those of other crops, although some species contain unique components, such as long-chain omega-3 fatty acids, not found in conventional crops. From the production/technological perspective, algae farming faces similar problems to those encountered in agriculture and aquaculture. Further, algaculture is not understood by most decision-makers, investors and other stakeholders, due to a lack of knowledge and background in this merging of agriculture with aquaculture.

As the fifth boundary, algae cultivation-based products are crops. Stakeholders lack specific knowledge of the requirements for their production.

### 7. What Is Marine Agronomy?

Marine agronomy refers to the science and technology of producing and using marine algae and marine aquatic plants for food, fuel, fibres and ocean restoration [17]. The main marine production crops are aquaculture macroalgae (seaweed). Most seaweeds produced in Asia are a result of marine agronomy or algaculture in the ocean (Figure 1), sometimes combined with fish and/or shellfish production with benefit from available organic matter. The seaweed production in Europe is still emerging but recently has exhibited a very high rate of growth.

"Seaweeding" is a relatively new term, in analogy to fishing, for harvesting wild seaweed (macroalgae) (Figure 2), relating to the harvesting of seaweed from the ocean either on-, near- or off-shore with nets (Figure 3). European seaweed industries that transform algae into alginates, carrageenan and other marine colloids or chemicals obtain their raw materials from wild seaweed resources algae sustainably harvested in the ocean [18]. However, seaweeding is not aquaculture, which requires a process where some form of intervention exists along the rearing process to enhance production, such as regular stocking, feeding and protection from predators, and also requires the individual or corporate ownership of the stock being cultivated.

As the sixth boundary, regulations to harvest seaweed from the ocean are often complex.

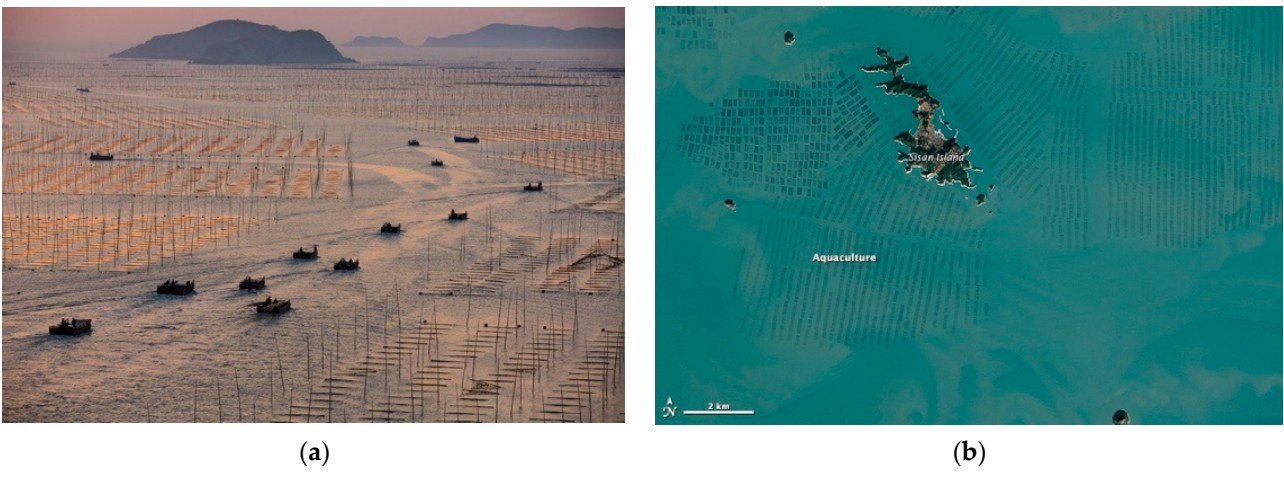

(**a**)　　　　　　　　　　　　　　　　(**b**)

**Figure 2.** (**a**) Seaweed farmers along the coast in Fujian, China. (**b**) Seaweed and oyster aquaculture on Sisan Island in South Korea.

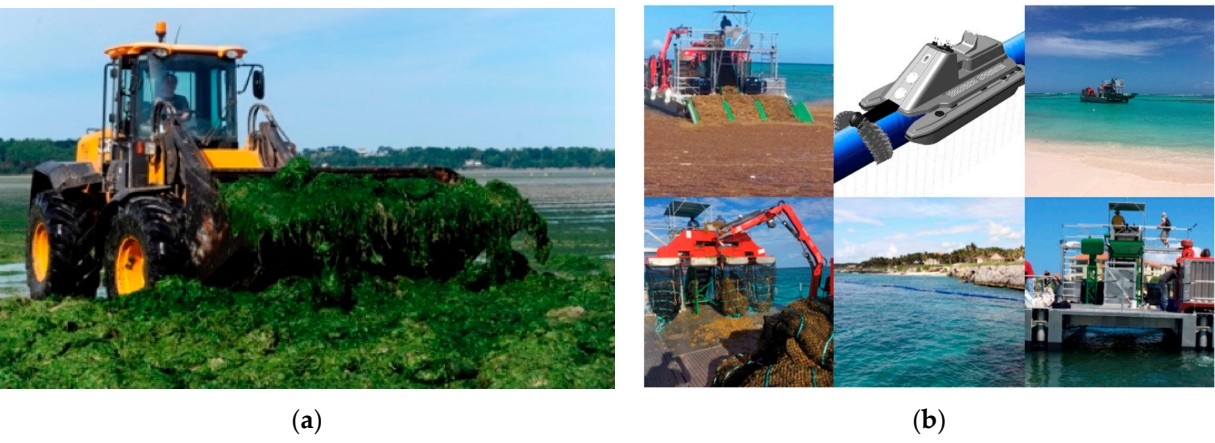

(**a**)　　　　　　　　　　　　　　　　(**b**)

**Figure 3.** (**a**) Seaweed harvesting in Britany, France. (**b**) Sargassum harvesting by Algenova in the Dominican Republic.

### 8. What Is the Algae Industry?

The algae industry produces and harvests algal biomass and then transforms it into products, using both mechanical and chemical operations.

> Are algal pastes or powders from natural harvest industrial products?

No. Algae paste and algae powders derived from harvesting seaweeds or microalgae blooms are not industrial products, because there is no production or transformation process involved.

> Is algal biomass cultivated in ponds, photobioreactors or fermenters an industrial product?

Yes. Algae production in reactors requires specific production processes and therefore are industrial products. Microalgae production is analogous to bacteria and yeast production at an industrial scale, with industrial processes ranging from closed dark fermentations to indoor autotrophic production and large-scale outdoor ponds.

> Is the production of algae extracts an industrial process?

Yes. The production of an extract usually requires a disruption of the biomass and the use of a chemical or mechanical extraction process and is, therefore, an industrial process.

As the seventh boundary, an algal industrial process requires the production of the algal biomass or the transformation of algae harvested from natural environments into a product different from the whole biomass.

### 9. How Is Algaculture Compared with Aquaculture and Agriculture?

Algaculture is complex, as it combines the knowledge of biology and engineering from both agriculture and aquaculture (Table 1), and is a relatively new sector when compared with agriculture and aquaculture: The Google index for agriculture is 870 million; for aquaculture, it is 38 million, and for algaeculture, it is only 413.000. There is a long way to go.

As the eighth boundary, algaculture is not a globally established and recognized sector.

**Table 1.** Major differences between agriculture, algaculture and aquaculture.

|  | Agriculture | Algaculture | Aquaculture |
|---|---|---|---|
| what | Multicellular plants with roots | Simple aquatic organisms without roots | Evolved organisms that live in water |
| when | Production started 10,000 years BC | Seaweed: production started in the 15th century Microalgae: production started in the 20th century | Production started 4000 years BC |
| how | Hydroponics > | Marine and freshwater | <Aquaponics |
| quantity | Wheat: 750 million metric tons Soy: 360 million metric tons | Seaweed: 17.3 million metric tons Microalgae: 50.000 tons (estimation) | Salmon: 1.4 million tons Aquaculture production: 171 million tons |
| scale | Mudanjiang (China): 9,105,426 ha South Dakotans (USA): 225,000 ha | The largest seaweed: Jiaozhou Bay: 2400 ha The largest spirulina farms: 40–50 hectares (USA, China) | Open-ocean in Panama: openblue 3 kton/year In-land farm in Saudi Arabia: Naqua 15,000 ton shrimp/year (4.177 ha) |

### 10. What Is Algae Biomass?

The term "algae biomass" is used, when there is cultivation, harvesting, stabilization or other processing that results in an algae product. In the case of either seaweed or microalgae, because they grow in the water, a drying process is usually required. Microalgae also require specific harvesting technologies because of their microscopic size. The drying of microalgae is industrially carried out, mainly using spray-dryers, and it is performed to reduce the water content to 5% or below. For macroalgae, sun or air drying is typically used. Other

stabilization methods, such as ensiling, are also possible. These processes produce stable algal biomass that then can be converted into foods, feeds and other products.

As the ninth boundary, the harvesting and stabilization of algal biomass is the first step to producing useful products.

## 11. What Are Organic Algae?

Organic refers to products that meet certain regulatory requirements such as cultivation without synthetic fertilizers pesticides and other chemicals, the use of genetically modified organisms (GMOs) or wastewaters. Regulations are set in the USA by the US Department of Agriculture under Organics Food Production Act and in Europe under a similar legal and regulatory framework, with both US and EU organic regulations generally mutually compatible to have an "organic" certification. The final product needs to meet the requirements of EU Regulation 2018/848 on organic production and the labelling of organic products. Euroleaf is the EU logo that identifies packaged organic food products [19]. Organic regulations designed around soil-based systems do not transfer well into algaculture or aquaculture. In addition, organic seaweed regulation cannot be used for microalgae. The main difference is that in organic agriculture organic fertilizers are applied in the soil which are not in direct contact with the crops (e.g., corn and fruits). Finding suitable organic fertilizers for algaculture is a challenge.

> Does organic microalgae cultivation require a "pasteurized" organic carbon source?

No. There are no specific requirements for the use of pasteurized or sterilized ingredients for the organic production of algae. However, good practices must be followed to avoid the potential contamination of the final products.

> Are seaweeds always organic products?

No. Specific regulations define the inputs and productions modes to achieve organic production status.

As the tenth boundary, organic algae require specific aquaculture-based regulations.

## 12. Can Algae Grown in Wastewater Be Considered as Algal Biomass?

Yes. Wastewaters, either industrial or urban (and including from agriculture, animal farming and also aquaculture), contain nutrients, such as nitrates, phosphates, microelements and organic substrates. The biomass grown in such substrates under sunlight is a mix of mainly algae and bacteria, with fungi and other organisms present. That resulting biomass can only be considered in the production of products that do not directly enter the human food chain, such as biofuels, biofertilizers and bioplastics. Only if safety and quality are assured, such as in the treatment of some food processing, agricultural or aquacultural wastes, could wastewater grown biomass be considered in the production of animal feeds. However, in general, in wastewater treatment processes, the resulting algal-bacterial product is not greatly different from the bacterial biomass produced in, for example, the activated sludge from conventional wastewater treatment [20].

> Can algae produced from wastewater be applied directly or indirectly for foods or feeds?

No. Existing regulations prevent the use of materials, algal or not, recovered from or related to wastewaters for human uses. This is a prevention measure to ensure the health of the population. It can be noted that "toilet to tap" systems are becoming increasingly necessary to deal with the increasing water shortages, suggesting a future pathway to the integration of wastewater treatment into the circular economy.

> Can algae produced from wastewater be in animal feeds, including aquaculture?

No, except in some cases. Wastewater contains useful nutrients for animal farming either urban and industrial. To prevent potential health problems, it is forbidden to use algae biomass produced on urban wastewater in animal feeds. However, manure is accepted as a raw material for some aquaculture applications, although strict regulations exist regarding the certification of the safety of material to be used, including chemical/biochemical

composition, microbiological contamination, heavy metals and other toxic compounds contents.

> Can algae produced from wastewater be applied in agriculture as biofertilizers or biostimulants?

Yes. Algae biomass grown on wastewaters contains useful fertilizers and, sometimes, plants growth stimulators. Thus, algae biomass produced in wastewater can be used for agriculture uses, regulated to avoid chemical, biochemical and microbiological, toxins contamination.

> Can algae biomass from wastewater be used for biofuels, bioplastics or chemicals?

Yes. Potentially, algae biomass produced during wastewater treatment processes is the most suitable for biofuels production due to its low production cost and availability, and stabilizing the biomass is needed. Possible routes to transform the algae biomass into biofuels include the following: (i) anaerobic digestion to produce biogas (methane + carbon dioxide) by methanogenic bacteria or alcoholic fermentation using selected yeast or bacteria; (ii) chemical or enzymatic transesterification of the algal lipids to biodiesel; and (iii) thermal processes such as pyrolysis or hydrothermal liquefaction to produce "green crude". The final upgrading, recovery and refining steps are required to meet specific fuel products and regulations.

As the eleventh boundary, when using wastewaters to produce algae there are regulatory and technical issues, limiting the application of produced biomass to mainly biofuels, biofertilizers and bioplastics.

### 13. What Are Toxic Algae and Harmful Algal Blooms?

Some species of algae produce toxins to compete with other algae or deter grazers and other predators. Often toxic algae bloom in a natural environment and, even if not toxic, can be harmful to the ecosystems by depleting oxygen when the bloom decays, resulting in "dead zones" with fish kills, noxious odors and foul taste of water supplies. Although the vast majority of algae, including bloom formers, are not toxic, some harmful algal blooms are associated with algae-produced toxins [21]. Harmful algal blooms can last from a few days to many months and are mostly caused by eutrophication—an overabundance of nutrients, mainly fixed nitrogen (nitrates, ammonia, etc.) and phosphates in the water, released from human settlements and agriculture. High water temperature and low circulation are contributing factors.

Harmful, sometimes toxic, algae blooms are caused by a few species of cyanobacteria, dinoflagellates and diatoms. Toxic algae can now be identified with multi-probe RNA chips and microarrays which are becoming a near-standard tool for toxin detection in algae blooms [22].

As the twelfth boundary, toxic microalgae and harmful algal blooms have significant economic impacts.

### 14. What Are the Major Bottlenecks for the Algaculture Scale-Up?

From the above discussion of the algae biomass sector, Table 2 summarizes 10 major issues for the major bottlenecks in algaculture. The specific ranking of weight or importance depends on who, where and when, and the overall impact and relevance depend on these factors.

The bottlenecks are interrelated, solving any one or even some, might not have sufficient impact to achieve the economic goal, and they must all be addressed and tackled in parallel and simultaneously. Logistics and locations for the large-scale deployment of algal production technologies and infrastructures is an immediate issue. Contaminations are related to crop protection on an ever-present challenge in algaculture. The market demand is often the limiting factor for scale-up, thus achieving economics of scale. The value chain is related to the lack of established products that can support the market demand. A "people bottleneck" is due to the lack of experienced professionals able to manage such production facilities. Investability is related to the combination of risk and

returns on investment. The product is related to the development and standardization of the final products. The business model is the way to achieve the scale-up. Bioprocessing is the necessary transformations between the production of biomass and the final products. Finally, politics drives the incentives and reduces the barriers to algae businesses, within a wider economic, social and environmental framework.

As the thirteenth boundary, the scale-up is the key bottleneck for the algae biomass sector and requires knowledge management.

**Table 2.** Most relevant bottlenecks for both macro- and microalgae.

| The Bottlenecks | What? | For Macroalgae (Land Off-Shore) | For Microalgae (Earth Lined Ponds) |
| --- | --- | --- | --- |
| 01. Logistics | Agriculture and aquaculture combined | Harsh marine environments | Complex infrastructures |
| 02. Contaminations | Crop protection and control | Difficult to control and predators | Wide-range, microscopic and unknown |
| 03. Market demand | Lack of market knowledge and valuation | The market is still very specific. | The production cost limits the market. |
| 04. Value chain | Complex and non-focused | Long and non-specialized | Long and non-specialized |
| 05. People | Lack of trained multi-specialists | Lack of seedling knowledge | Lack of experienced managers |
| 06. Investability | Reduced investment attractiveness | Long return on investment (*) | Very long return on investment |
| 07. Produtech | Final product formulation complexity | Processing-required products | Non-evident, high-value and small-scale |
| 08. Business model | Profitability at the small scale, before at the large scale | Relevant value only in complex business models | Relevant value only in complex business models |
| 09. Bioprocessing | Processing is required in the value chain. | Long processing from farm to fork | Long processing, from farm to fork |
| 10. Political incentives | Framework before sustainability | Global strategic impact of the Integrated multitrophic aquaculture | Advanced crop with low $CO_2$ emission |

(*) Unless for a very large scale.

## 15. How Is Microalgae Biodiversity Ubiquitous?

Algae are ubiquitous in marine, freshwater and terrestrial habitats, providing a broad biochemical diversity. The numbers of different algal microalgae species are not easily estimated, and over ten thousand have been described, with some hundreds thousand mating types estimated only for some species of diatoms. Possibly, a million or more species may exist, depending on what definition of species is adopted. The different species of algae cover all water bodies and most land areas on the planet, including extreme environments from ice (such as *Chlamydomonas nivalis* and relatives) to hot springs (such as *Mastigocladus laminosus* that survives up to 64 °C) and hypersaline environments (such as *Dunaliella salina*). It was shown that in the species of *Dunaliella salina*, the same strains exist in different locations across the world [23]. *Lobosphaera incisa* (previously known as *Parietochloris incisa*) isolated from Mount Tateyama, Japan is genetically identical to the same species isolated in a water lake from a mine in the Czech Republic.

The European Algae Biomass Association has catalogued 75 microalgae culture collections worldwide, and each of them has between 500 and 1000 different strains. However, it is estimated by the culture collection curators that 20% to 30% of the registered species are wrongly classified; and more than half the species are common to most collections even if isolated in a different location. The emergence of DNA barcoding and other molecular biology techniques will bring new light to the algae taxonomy, and therefore, we will see changes of names at both the species and genus levels. More than 70,000 different algae

species are known, and they will undergo a genetic-based reorganization over the next 10 years [24].

> Can algae, either macro- or microalgae, have industrial Property protection, like patents?

No and Yes. It is not possible to patent living organisms already existing on the planet. However, it is possible to obtain patents for strains bred or genetically modified to yield novel phenotypes. In the US, it is possible to apply for a "plant patent" for an asexually reproduced and distinctive new plant variety. It is also possible to patent and has IP protection for GMOs and related processes. The Nagoya protocol also provides geographical recognition, and in some cases, benefits can be provided for the location of origin. The Material Transfer Agreement (MTA) is a contract that governs the transfer of tangible research materials between two and defines the rights of the provider and the recipient for the materials and any derivatives. (www.wipo.int/tk/en/databases/contracts/texts/bio.html; accessed on 25 January 2021). It is also possible to have patents about the production and processing of natural living organisms for specific applications.

As the fourteenth boundary, algae species, especially microalgae, are biodiverse and ubiquitous with the same species present in different geographies. Similar ecological niches worldwide show the same microalgae strains.

## 16. Conclusions

It is difficult to straightforwardly answer all questions raised above, as the "reality" is complex. However, the understanding of boundaries can help to separate "yes" and "no". Innovation bottlenecks result, when lacking the knowledge from basic processes to regulatory issues, hinders the development and scale-up of new products and processes. The innovation potential of the algae biomass sector is due to algae being compatible with agriculture and aquaculture and is related industries and services. Algae are highly diverse in size, from *Ostreococcus tauri*, a picoplankton that is less than one micron, to the gigantic Pacific kelp, *Macrocystis pyrifera*, which can grow up to 60 m. Microalgae are also highly diverse in their habitats, from freshwater to hypersaline and high-temperature environments, for example, and can grow in autotrophic, heterotrophic or mixotrophic modes. Seaweeds, by contrast, only grow in ocean habitats with narrow salinity and temperature requirements. Algae have great application potential in largely different fields including the following: (i) human nutrition and health (pharma, cosmetic, nutraceuticals and foods); (ii) foods production (aquaculture, animal feeding and agriculture); (iii) materials and energy (chemical commodities, bioplastics and biofuels); and (iv) bioremediation ($CO_2$ capture from flue gases, soil regeneration and wastewater treatment). All these applications and production frameworks not only provide a high potential for algae production, but also entail complex and extensive regulatory frameworks. The above discussions suggest that the boundaries and the bottlenecks for innovation in algaculture can be overcome, suggesting major advances in this field.

**Author Contributions:** Conceptualization, V.V.V., J.-P.C.; writing—original draft preparation, V.V.V., J.-P.C.; writing—review and editing, F.G.A., J.B. All authors have read and agreed to the published version of the manuscript.

**Funding:** This work is part of the DigitAlgaesation project (A knowledge-based training network for digitalisation of photosynthetic bioprocesses) Grant Agreement Number 955520.

**Institutional Review Board Statement:** Not applicable.

**Informed Consent Statement:** Not applicable.

**Data Availability Statement:** Not applicable.

**Conflicts of Interest:** The authors declare no conflict of interest.

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
