# Peer review of "Clarification of Most Relevant Concepts Related to the Microalgae Production Sector"

_processes, doi:10.3390/pr10010175_

Round 1

Reviewer 1 Report

Innovation bottlenecks for the microalgae production sector related to unclear understanding about regulatory boundaries

 By Vítor Verdelho Vieira et al.

The key topics related to understanding regulatory boundaries concerning algae have been dealt with. The subject of this article is a long time required discussion. Key topics related to algae are well presented and discussed in a question-and-answer format.

Some flaws of the article are highlighted in the following points:

Page 1 - Correct Europrean to European

Language improvement is required.

Introduction

The intent of the article as presented in the introduction is to deal with microalgae. But in the text, the questions mainly refer to algae. This is confusing.  Among the microalgae the genus Spirulina is included. Thus, the reader understands that the authors include the cyanobacteria in the microalgae. Is this what the authors wish?

Line 75.  Despite the fact that in the introduction only microalgae are mentioned: the first question is: what are algae?

Line 78 –“Converting CO2 to biomass and O2”

In oxygenic photosynthesis, oxygen derives from water not from CO2

Line 92 and line 100 “Algae can be grouped into macroalgae or seaweeds (macroscopic multicellular species free-floating, sometimes attached to the bottom of seas, rivers and lakes), and microalgae (microscopic, unicellular filamentous or colonial species called microalgae)”

Cyanobacteria are not mentioned. Where cyanobacteria, although included in the term algae, are dealt with? Why is not mentioned that they are prokaryotic and thus very diverse from microalgae?

 In line 104 the question is > Are algae aquatic organisms?

In line 105 the answer is Yes. Generally, microalgae are aquatic organisms.

Again, despite the question is on algae, the answer is on microalgae.

Line 114 and 115-

“Microalgae were the responsible of current life on the planet. Early in the history of life, algae changed the planet’s atmosphere by producing oxygen, thus paving the way for the evolution of eukaryotic organisms, including us.”

This statement is misleading. The CYANOBACTERIA were the responsible for the oxygenation of the primordial atmosphere.  

ERROR: algae cannot pave the way for the evolution of eukaryotic organisms. Excluding cyanobacteria, they are already eukaryotic.

 Line 118-

“Microalgae as single-cell organisms can replicate in a few hours, making it possible to produce large amounts quickly”.

Some microalgae are not single celled organisms

Not all the microalgae can replicate in a few hours

Line 174 –

“Algae biomass is not globally different from other crops, thus its biochemical composition and nutritional properties are analogous to other crops”.

Are the authors really convinced of this statement?

Line 220 - 221

> Algae cultivation in fermentors is an industrial process?

“No. Algae production in fermentors requires a specific aquaculture device and therefore it is included in the primary sector. The fermentor device is just where the aquatic organisms multiply. The vessel used for the production is different from conventional aquaculture”

Fermentors are treated in books of industrial microbiology. The cultivation of yeast or bacteria in fermentors is industrial microbiology. What is the reason why the cultivation of microalgae in fermentors is not?

Line 252-

“The harvesting of algae is the same process as using nets to catch fish - but algae are not centimeter size but micro”.

Are the authors forgetting the macroalgae?

Author Response

Reviewer 1

The key topics related to understanding regulatory boundaries concerning algae have been dealt with. The subject of this article is a long time required discussion. Key topics related to algae are well presented and discussed in a question-and-answer format.

Response: Thanks for the positive comment from the reviewer.

Some flaws of the article are highlighted in the following points:

Page 1 - Correct Europrean to European. Language improvement is required.

Response: this word has been revised, the entire document being reviewed by native speakers.

Introduction. The intent of the article as presented in the introduction is to deal with microalgae. But in the text, the questions mainly refer to algae. This is confusing.  Among the microalgae the genus Spirulina is included. Thus, the reader understands that the authors include the cyanobacteria in the microalgae. Is this what the authors wish?

Response: Yes. This question has been clarified from the beginning by including a sentence at the abstract.

Line 75.  Despite the fact that in the introduction only microalgae are mentioned: the first question is: what are algae?

Response: a sentence has been added at the beginning of the introduction to clarify that the term microalgae also include cyanobacteria.

Line 78 –“Converting CO2 to biomass and O2” In oxygenic photosynthesis, oxygen derives from water not from CO2

Response: Yes, it has been clarified.

Line 92 and line 100 “Algae can be grouped into macroalgae or seaweeds (macroscopic multicellular species free-floating, sometimes attached to the bottom of seas, rivers and lakes), and microalgae (microscopic, unicellular filamentous or colonial species called microalgae)”. Cyanobacteria are not mentioned. Where cyanobacteria, although included in the term algae, are dealt with? Why is not mentioned that they are prokaryotic and thus very diverse from microalgae?

Response: Yes, a sentence has been included to clarify this point.

In line 104 the question is > Are algae aquatic organisms? In line 105 the answer is Yes. Generally, microalgae are aquatic organisms. Again, despite the question is on algae, the answer is on microalgae.

Response: In the new version of the manuscript it is explained that the term microalgae include both eukaryotic microalgae and prokaryotic cyanobacteria

Line 114 and 115-“Microalgae were the responsible of current life on the planet. Early in the history of life, algae changed the planet’s atmosphere by producing oxygen, thus paving the way for the evolution of eukaryotic organisms, including us.” This statement is misleading. The CYANOBACTERIA were the responsible for the oxygenation of the primordial atmosphere.  ERROR: algae cannot pave the way for the evolution of eukaryotic organisms. Excluding cyanobacteria, they are already eukaryotic.

Response: This sentence has been rewritten according to the reviewer comment.

Line 118-“Microalgae as single-cell organisms can replicate in a few hours, making it possible to produce large amounts quickly”. Some microalgae are not single-celled organisms. Not all microalgae can replicate in a few hours.

Response: This sentence has been rewritten to include the comment from the reviewer.

Line 174 –“Algae biomass is not globally different from other crops, thus its biochemical composition and nutritional properties are analogous to other crops”. Are the authors really convinced of this statement?

Response: In general algae biomass is analogous to other biomasses. Of course, some differences exist between algae and other biomasses such as cellulosic, animal, cereals, etc.. but major components are analogous.

Line 220 – 221 > Algae cultivation in fermentors is an industrial process? “No. Algae production in fermentors requires a specific aquaculture device and therefore it is included in the primary sector. The fermentor device is just where the aquatic organisms multiply. The vessel used for the production is different from conventional aquaculture”. Fermentors are treated in books of industrial microbiology. The cultivation of yeast or bacteria in fermentors is industrial microbiology. What is the reason why the cultivation of microalgae in fermentors is not?

Response: agree with the reviewer this sentence has been rewritten.

Line 252-“The harvesting of algae is the same process as using nets to catch fish - but algae are not centimeter size but micro”. Are the authors forgetting the macroalgae?

Response: agree with the reviewer this sentence has been rewritten.

Reviewer 2 Report

Dear author(s),
this manuscript brings some inspiring insights to algae management and I tend to quickly agree on its publication. However, there are some issues that can be immediately addressed to improve the overall communication of your work:

Title:

  • condensate the main discovery into a short claim (full sentence, don't omit the verb)
  • shortening advisable

Abstract:

  • follow the established schema: A/ motivation + research hypothesis; B/ methods + results; C/ conclusions and interdisciplinary implications
  • it is hard to find something new or unexpected (provide answers to all questions provided), highlight the scientific novelty and quantify the economic importance of your discovery, clarify how will humanity benefit from your work
  • avoid typos like "??"

Introduction:

  • remove reference overkill by breaking down all clusters of references (use only 1 reference per claim/sentence)
  • deeper review the latest findings regarding interactions of algae and economy/industry, refer to papers "Economic Considerations on Nutrient Utilization in Wastewater Management" and "Deep Learning-enabled Smart Process Planning in Cyber-Physical System-based Manufacturing"
  • write more technically (always provide corresponding numbers) and be more straightforward (condensate your text, remove ballast phrases and cliche)
  • clearly build your research hypothesis at the end of the Introduction chapter, better justify its urgency and significance (exclude those research questions whose urgency or importance is not essential)

Methodology:

  • in this chapter, our readers should find here (only and exclusively) detailed description of all your procedures (step by step, describe each apparatus and method used), anybody who reads this chapter should be able to repeat your methods and obtain exactly the same result
  • indicate the research databases used, the Keywords searched, time span etc.

Results and Discussion:

  • all limitations of your methods and results should be critically discussed; the results should be statistically evaluated and compared with the previous state of knowledge, the differences should be justified and subjected to synthesis which reveals new theoretical findings
  • discuss papers that deals with modern manufacturing issues, refer to papers "Networked, Smart, and Responsive Devices in Industry 4.0 Manufacturing Systems" and "Green Entrepreneurship: Literature Review and Agenda for Future Research"
  • each Fig. and Tab. should be provided with detailed caption that will explain A/ what can be seen; B/ why is it important and C/ how is it related to the research hypothesis
  •  take your research to the next level, provide a deeper synthesis of your results and reveal the mechanisms that shape them, this will allow you to uncover original theoretical insights

Conclusions:

  • conclusion is not the same as summary of your work (do not repeat your methods and results again and again), make sure you are presenting only new theoretical findings that originate firstly from your work and are not deducible from other literature
  • clearly indicate whether your research hypothesis tends to be confirmed or not, highlight/quantify the industrial/environmental significance

Author Response

Reviewer 2

This manuscript brings some inspiring insights to algae management and I tend to quickly agree with its publication. However, some issues can be immediately addressed to improve the overall communication of your work:

Title: condensate the main discovery into a short claim (full sentence, don't omit the verb) shortening advisable

Response: Agree with the reviewer the title has been revised.

Abstract: follow the established schema: A/ motivation + research hypothesis; B/ methods + results; C/ conclusions and interdisciplinary implications. it is hard to find something new or unexpected (provide answers to all questions provided), highlight the scientific novelty and quantify the economic importance of your discovery, clarify how will humanity benefit from your work.

Response: This paper is not a regular research paper, then it is difficult to provide a conventional scheme about new phenomena or data provided. Otherwise, the paper provides a rationale of relevant concepts related to microalgae production, especially to fix unclear understanding of major concepts.

avoid typos like "??"

Response: This typo has been removed.

Introduction: remove reference overkill by breaking down all clusters of references (use only 1 reference per claim/sentence)- deeper review the latest findings regarding interactions of algae and economy/industry, refer to papers "Economic Considerations on Nutrient Utilization in Wastewater Management" and "Deep Learning-enabled Smart Process Planning in Cyber-Physical System-based Manufacturing". write more technically (always provide corresponding numbers) and be more straightforward (condensate your text, remove ballast phrases and cliche). clearly build your research hypothesis at the end of the Introduction chapter, better justify its urgency and significance (exclude those research questions whose urgency or importance is not essential).

Response: agree with the reviewer the list of references has been revised, removing duplicates and including new ones needed. Recommended sentences have been included. Also, the aim of the paper is already included at the end of the introduction section.

Methodology: In this chapter, our readers should find here (only and exclusively) detailed description of all your procedures (step by step, describe each apparatus and method used), anybody who reads this chapter should be able to repeat your methods and obtain exactly the same result indicate the research databases used, the Keywords searched, time span etc.

Response: In regular papers methodology is aimed to shows the methods and analytics already utilized. In this case, because this is more an opinion paper this section is not needed.

Results and Discussion: all limitations of your methods and results should be critically discussed; the results should be statistically evaluated and compared with the previous state of knowledge, the differences should be justified and subjected to synthesis which reveals new theoretical findings discuss papers that deals with modern manufacturing issues, refer to papers "Networked, Smart, and Responsive Devices in Industry 4.0 Manufacturing Systems" and "Green Entrepreneurship: Literature Review and Agenda for Future Research" each Fig. and Tab. should be provided with detailed caption that will explain A/ what can be seen; B/ why is it important and C/ how is it related to the research hypothesis  take your research to the next level, provide a deeper synthesis of your results and reveal the mechanisms that shape them, this will allow you to uncover original theoretical insights

Response: Again this paper is not related to new data or phenomena, but to provide a clear description of the most relevant concepts related to microalgae production, including cyanobacteria. Figures already included have adequate figure captions and shows relevant aspects of algae production and processing.

Conclusions: conclusion is not the same as summary of your work (do not repeat your methods and results again and again), make sure you are presenting only new theoretical findings that originate firstly from your work and are not deducible from other literature clearly indicate whether your research hypothesis tends to be confirmed or not, highlight/quantify the industrial/environmental significance.

Response: The conclusions section is different from the abstract. Thus, the rationale of algae production processes and the most relevant aspects related to them are included.

Reviewer 3 Report

Dear authors,

I was very much intrigued and eager to learn about the innovation bottlenecks for microalgae production sector, but your communication fails to meet the expectations seeded in the title.

There are two main reasons for rejecting this paper:

  1. The focus is lost over the paper, indicating that it will focus on microalgae but than there are sections that strictly address macroalgae (e.g. 6th and 7th boundary). The bottlenecks highlighted are holistic to any novel biomass supply in the emerging bio-based industry, called market failure – when industry demand does not meet the biomass supply for any reasons. Although there are solutions for those challenges from other sectors, there are not suggested in this paper. Conclusions fail to meet the research question suggested from the title. I propose that revisiting the more detailed comments (below), together with more support from the existing knowledge (citations) would improve the paper to be considered for publishing.
  2. Lack of structure to deliver a message: I am aware that MDPI allows more flexible structure for reviews and other papers than original scientific papers defined in Manuscript template, but when comparing the structure of your paper to those that are published, they vary significantly. Please check (randomly chosen) communication published in MDPI Processes:

https://www.mdpi.com/2227-9717/9/1/58

https://www.mdpi.com/2227-9717/8/10/1263/htm

https://www.mdpi.com/2227-9717/8/6/688/htm

Citations are not properly cited, neither as references nor in the text. I strongly advise you to reorganise your work accordingly in the next attempt to publish.

Good luck in your future endeavours!

More detailed comments:

  1. Adjust Abstract according to the text alterations to be made
  2. Introduction:
    1. I suggest making the first paragraph shorter and more factual. For instance: Given the wide range of possible applications in the bio-based industry, varying from (i) (…lines39-40…), and ability to achieve carbon sinks, lots of expectation is placed on microalgae industry. Yet, commercial processes (…) continue with line 42.
    2. Line 42 contradicts with line 72.
    3. Line 47 price represents a range, thus “respectively” can be used in such statement. Please revise.
    4. Line 47 price contradicts with the lines 52 and 54.
    5. Line 62 add “ and business models” after “schemes”.
    6. Line 63: add “finding new value chains to improve circularity and material efficiency” or similar, after comma

L95-99: shorten the text

L100-103: merge to L 94

L 110-111 why would that be a boundary?

L119.122: repetition

L183-207 not relevant for microalgae, please check

L216: please check the definition of industrial product before having the claim: https://notesmatic.com/industrial-products-types-and-characteristics/ for instance

Wheat flour is considered as an industrial product, there should be analogy with algae powder.

L219: can you make any analogy with more advanced biotechnical activities cited in the Introduction?

L235: table captions are not readable, not in template format. “aquaculture production”, under “salmon”: what kind of produce?; check unit after 15.000.

L237: table captions are not readable, not in template format.

L262: why is that a boundary?

L288-294; repetition

L344: HAB instead of full name

L352-371: there is a vast effort and knowledge on scaling up innovations and processes that could be applied to this section. To my view, this is one of the weakest sections in the paper.

A general comment: maybe to list the bottlenecks with suggested solutions as a summary table?

  1. Conclusions need to be rewritten.

Author Response

Reviewer 3

I was very much intrigued and eager to learn about the innovation bottlenecks for microalgae production sector, but your communication fails to meet the expectations seeded in the title.

Response: Agree with the reviewer the title has been modified to better fit the content of the work.

There are two main reasons for rejecting this paper:

  1. The focus is lost over the paper, indicating that it will focus on microalgae but than there are sections that strictly address macroalgae (e.g. 6th and 7th boundary). The bottlenecks highlighted are holistic to any novel biomass supply in the emerging bio-based industry, called market failure – when industry demand does not meet the biomass supply for any reasons. Although there are solutions for those challenges from other sectors, there are not suggested in this paper. Conclusions fail to meet the research question suggested from the title. I propose that revisiting the more detailed comments (below), together with more support from the existing knowledge (citations) would improve the paper to be considered for publishing.

Response: Agree with the reviewer the paper includes macroalgae related sentences because both macro and microalgae sectors are closely related. Also to distiguish between both applications is needed. The aim of the paper is not to provide a list of limiting factors and potential solutions but to clarify some terms and definitions needed to avoid mistakes and problems already showed both in papers and conferences. Because the paper is supported by the EABA we consider this paper relevant for the sector and potentially useful for new investors and researchers in this field.

  1. Lack of structure to deliver a message: I am aware that MDPI allows more flexible structure for reviews and other papers than original scientific papers defined in Manuscript template, but when comparing the structure of your paper to those that are published, they vary significantly. Please check (randomly chosen) communication published in MDPI Processes: https://www.mdpi.com/2227-9717/9/1/58, https://www.mdpi.com/2227-9717/8/10/1263/htm, https://www.mdpi.com/2227-9717/8/6/688/htm

Response: Agree with the reviewer this paper has a different structure to regular research papers. This is more an opinion paper devoted to clarifying relevant aspects related to microalgae production. This paper aims to avoid the mistakes of new investors and researchers in this field, thus providing clear definitions and explanations of the most relevant aspects of this field. We consider that these papers, in addition to other similar papers, although have a different structure they are highly valuable to the field, which at the end must be the aim of whatever paper or journal.

Citations are not properly cited, neither as references nor in the text. I strongly advise you to reorganise your work accordingly in the next attempt to publish.

Response: Sorry, I don’t know the precise style for references on this journal, but because we use Mendeley we can adjust the format to whatever style required.

More detailed comments:

  1. Adjust Abstract according to the text alterations to be made

Response: done

  1. Introduction:
    1. I suggest making the first paragraph shorter and more factual. For instance: Given the wide range of possible applications in the bio-based industry, varying from (i) (…lines39-40…), and ability to achieve carbon sinks, lots of expectation is placed on microalgae industry. Yet, commercial processes (…) continue with line 42.

Response: On this first paragraph a complete overview of microalgae production applications and processes is provided. We consider it adequate.

    1. Line 42 contradicts with line 72.

Response: Sorry, I can not find this contradiction.

    1. Line 47 price represents a range, thus “respectively” can be used in such statement. Please revise.

Response: sorry, I didn’t find this error.

    1. Line 47 price contradicts with the lines 52 and 54.

Response: sorry, I didn’t find this error.

    1. Line 62 add “ and business models” after “schemes”.

Response: Done

    1. Line 63: add “finding new value chains to improve circularity and material efficiency” or similar, after comma

Response: Done

L95-99: shorten the text

Response: sorry, I didn’t find this error.

L100-103: merge to L 94

Response: sorry, I didn’t find this error.

L 110-111 why would that be a boundary?

Response: Boundary mean a limit for the meaning of the term.

L119.122: repetition.

Response: corrected.

L183-207 not relevant for microalgae, please check

Response: sorry, I didn’t find this error.

L216: please check the definition of industrial product before having the claim: https://notesmatic.com/industrial-products-types-and-characteristics/ for instance Wheat flour is considered as an industrial product, there should be analogy with algae powder.

Response: Because in algae biomass production operations such as milling, desintegration or extraction are not performed the production of algae powder after spray-dryer is not industrial.

L219: can you make any analogy with more advanced biotechnical activities cited in the Introduction?

Response: Done

L235: table captions are not readable, not in template format. “aquaculture production”, under “salmon”: what kind of produce?; check unit after 15.000.

Response: The caption of table has been revised, also the units.

L237: table captions are not readable, not in template format.

Response: The caption of entire tables and figures have been revised.

L262: why is that a boundary?

Response: Boundary means that this is a relevant factor to be considered.

L288-294; repetition

Response: the sentence has been revised.

L344: HAB instead of full name

Response: done

L352-371: there is a vast effort and knowledge on scaling up innovations and processes that could be applied to this section. To my view, this is one of the weakest sections in the paper.

Response: Agree to the reviewer a deeper analysis of potential innovations could be included. However, this is not the aim of the paper, it being more devoted to clarifying concepts and avoiding mistakes by new entrepreneurs and young researchers.

A general comment: maybe to list the bottlenecks with suggested solutions as a summary table?

Response: Agree with the reviewer it would be useful for some other papers, but this is not the aim of this paper.

  1. Conclusions need to be rewritten.

Response: conclusions summarize the most relevant concepts related to microalgae production processes in comparison with similar fields such as macroalgae or aquaculture. This is the aim of the paper then it being adequate for this purpose.

Round 2

Reviewer 3 Report

Dear authors,

you have improved the paper but rejected most of the comments.

I don't see that opening of the paper " Microalgae, including cyanobacteria, have been proposed as Holy Grail to solve the problems of mankind such as food and biofuels supply in addition to greenhouse gases abatement, among others. " has scientific merit. I advise you to make it more factual and less dramatical.

You must have been editing your text with some other reviewer's comments and changed the number of lines. That is why you could not intervene. For instance, L 42 is now L52 and still has the same oversight: " the current biomass production cost of 5-20 €/kg when using open and closed systems respectively." If one uses a range (5-20 €/kg), it can't be related to "open and closed systems, respectively." Please, check: https://www.springer.com/gp/authors-editors/authorandreviewertutorials/writinginenglish/use-of-respectively/10252704.

Please, have a look at the original manuscript submitted to the journal and address the comments.

Thank you in advance!

Author Response

Response to the reviewer:

you have improved the paper but rejected most of the comments.

Response: We try to include all the suggestions from reviewers, but in some cases, we were not able to do it.

I don't see that opening of the paper " Microalgae, including cyanobacteria, have been proposed as Holy Grail to solve the problems of mankind such as food and biofuels supply in addition to greenhouse gases abatement, among others. " has scientific merit. I advise you to make it more factual and less dramatical.

Response: This sentence has been removed.

You must have been editing your text with some other reviewer's comments and changed the number of lines. That is why you could not intervene. For instance, L 42 is now L52 and still has the same oversight: " the current biomass production cost of 5-20 €/kg when using open and closed systems respectively." If one uses a range (5-20 €/kg), it can't be related to "open and closed systems, respectively." Please, check: https://www.springer.com/gp/authors-editors/authorandreviewertutorials/writinginenglish/use-of-respectively/10252704.

Response: This sentence has been revised.

Please, have a look at the original manuscript submitted to the journal and address the comments.

Response: The entire document has been reviewed by a native English speaker and well-recognized expert on the microalgae field.

Round 3

Reviewer 3 Report

Dear authors,

exactly for the reason that the paper is supported by the EABA, it will be very much read. We both agree that this paper relevant for the sector and potentially useful for new investors and researchers in this field.

However, I see that you haven't changed the first sentence although you claim you did. And I see that the referencing style of the Journal is not of your concern either. 

This is a 3rd round of revision where you decide not to reflect on the reviewer opinion which is actually targeted to improve your paper. 

I will let the Editors decide how to proceed.

Best of luck.

Author Response

Sorry, I don't know what sentence you mean. I already review the first sentence of both the abstract and introduction sections and I didn't find problems with that sentences. The style of the paper is really different but we think that in this way the paper can be more useful. I greatly appreciate your opinion and help to improve the quality of the paper. THANKS...